# Simulating the Laurentide ice sheet of the Last Glacial Maximum

Daniel Moreno-Parada[1,2], Jorge Alvarez-Solas[1,2], Javier Blasco[1,2,3], Marisa Montoya[1,2], and Alexander Robinson[1,2,4]

[1]Departamento de Física de la Tierra y Astrofísica, Universidad Complutense de Madrid, Facultad de Ciencias Físicas, 28040 Madrid, Spain
[2]Instituto de Geociencias, Consejo Superior de Investigaciones Cientifícas-Universidad Complutense de Madrid, 28040 Madrid, Spain
[3]Laboratoire de Glaciologie, Université Libre de Bruxelles, Brussels, Belgium
[4]Potsdam Institute for Climate Impact Research, 14473 Potsdam, Germany

**Correspondence:** Daniel Moreno Parada (danielm@ucm.es)

**Abstract.** In the last decades, great effort has been made to reconstruct the Laurentide Ice Sheet (LIS) during the Last Glacial Maximum (LGM, ca. 21,000 years before present, 21 kyr ago). Uncertainties underlying its modelling have led to notable differences in fundamental features such as its maximum elevation, extent and total volume. As a result, the uncertainty in ice dynamics and thus in ice extent, volume and ice-stream stability remains large. We herein use a higher-order three-dimensional
ice-sheet model to simulate the LIS under LGM boundary conditions for a number of basal friction formulations of varying complexity. Their consequences on the Laurentide ice streams, configuration, extent and volume are explicitly quantified. Total volume and ice extent generally reach a constant equilibrium value that falls close to prior LIS reconstructions. Simulations exhibit high sensitivity to the dependency of the basal shear stress on the sliding velocity. In particular, a regularized-Coulomb friction formulation appears to be the best choice in terms of ice volume and ice-stream realism. Pronounced differences are
found when the basal friction stress is thermomechanically coupled: the base remains colder and the LIS volume is lower than in the purely mechanical friction scenario counterpart. Thermomechanical coupling is fundamental for producing rapid ice streaming, yet it leads to a similar ice distribution overall.

## 1 Introduction

The Laurentide Ice Sheet (LIS) was the largest of the former Northern Hemisphere ice sheets during the Last Glacial Maximum (LGM, ca. 21,000 years before present, 21 kyr ago). The LIS may have advanced to its maximum extent as early as 29–27 kyr ago, well before the LGM, and remained near that limit until 17 kyr ago (Dyke et al., 2002; Tarasov et al., 2012). Consequently, the LIS was the main contributor to sea-level change during the last glacial period, with an estimated sea-level equivalent (SLE) of about 70 metres ($28 \times 10^6$ km$^3$) with respect to present (Peltier, 2004; Tarasov et al., 2012). Hereinafter, the LIS will refer
to the entire North American ice-sheet complex, i.e., including the Cordilleran, Innuitian and Laurentide ice sheets.

Great effort has been made to reconstruct the LIS at the LGM throughout the last five decades. Several approaches are found in the literature. The first numerical methods relied on simplified ice physics, a prescribed ice accumulation rate and ice surface temperature and the assumption that the ice sheet was in a steady state (e.g., Paterson, 1972; Sugden, 1977; Hughes et al., 1980). This assumption was later relaxed by Mahaffy (1976) and Jenssen (1977), though the model was not applied to the late glacial history of the LIS. A completely independent approach was taken by Clark (1980) based on an inversion study of sea-level data where none of the previous assumptions are applied. Strictly speaking, the inversion solely shared the ice extent with prior studies which is, in general, well known. This further allowed for independent tests of the reliability of such assumptions by comparison among ice sheets.

Reconstructions of the size and distribution of the LIS based on forward ice-sheet modelling at the LGM have long dealt with the implications of a heterogeneous bedrock geology on the ice-sheet flow dynamics (e.g., Calov et al., 2002; Tarasov and Peltier, 2004). The central core of the LIS rests on a hard bedrock of the Canadian shield whereas nearly the entire Hudson Bay and Hudson Strait consist of Paleozoic carbonates easily eroded into a soft, slippery base. In view of this configuration, two approaches were classically taken. First, a simplification of the bedrock complexity was made by ignoring this deformable bed, thus resulting in a single-domed reconstruction centred over Hudson Bay (Denton, 1981). The second approach considered lubricated basal conditions by reducing the maximum basal shear stress. Unlike the previous results, the reconstructions presented a multi-domed ice sheet with a thinner ice sheet and a less steep slope over Hudson Bay (Boulton et al., 1985; Fisher et al., 1985). This multi-domed configuration is also found in recent reconstructions (Tarasov et al., 2012; Gowan et al., 2021).

As a result of fundamental uncertainties underlying ice-sheet modelling of the LIS, its maximum elevation, extent and total volume differ largely among studies (Stokes, 2017). In particular, the total volume carries the greatest uncertainty. Originally, Ramsay (1931) estimated a total LIS volume of $45.45 \times 10^6$ km$^3$, with a $15.75 \times 10^6$ km$^2$ extent and a maximum elevation of 2.9 km (here, and subsequently, above present sea level). More than three decades later, Paterson (1972) provided a significantly lower volume estimation of $26.5 \times 10^6$ km$^3$ with $11.6 \times 10^6$ km$^2$ ice covered area and 2.7 km maximum ice thickness. The lowest overall volume estimate was given by Peltier (1994) (ICE-4G) with $19.0 \times 10^6$ km$^3$, whereas more recent studies yield $28 \times 10^6$ km$^3$ (Tarasov et al., 2012) and $35 \times 10^6$ km$^3$ (including the Cordilleran Ice Sheet, Gregoire et al., 2012).

Already noted by Clark (1980), the LIS may have never attained a steady state, and it was possibly a rather dynamic system with rapid variations of its southern margin as well as a variable Hudson Bay ice thickness. MacAyeal (1993a) later proposed a mechanism by which Hudson Bay would periodically switch from a surging to a purging state (controlling the flux of ice through Hudson Strait ice stream) and further tested his theoretical prediction with a simple model (MacAyeal, 1993b). In fact, the LIS mass loss is intimately related to a variable Hudson Bay ice thickness through rapidly-flowing ice streams that account for most of the ice sheet discharge (Stokes and Tarasov, 2010). Nevertheless, the representation of these ice streams into numerical ice-sheet models remains challenging. As a result, we lack a deeper comprehension of the role of ice streams which leads to larger model output uncertainties.

The reconstruction of paleo ice streams is typically based on two methods. The first one rests on the assumption that the subglacial imprint of streaming and non-streaming areas is distinct (e.g., Kleman et al., 1997; Stokes and Clark, 1999) and consists of gathering enough evidence from landforms and sediments so as to reproduce their dynamics (e.g., Winsborrow

et al., 2004; Ottesen et al., 2005). The second one is, again, based on forward ice-sheet modelling using numerical models capable of simulating ice streaming (e.g., Boulton and Hagdorn, 2006). This ability is usually provided by thermomechanical feedbacks in topographic troughs and parametrizations of ice-bed coupling strength over soft sediments (Marshall et al., 1996).

Despite the comprehensive work carried out in the last decades, none of these studies addressed the repercussions of different basal friction formulations when simulating the LIS during the LGM nor their explicit implications in ice extent, volume and ice-stream representation. In fact, recent studies have shown significant consequences of this uncertainty for the Antarctic Ice Sheet (e.g., Blasco et al., 2021). We herein consider three scenarios of varying dynamic complexity and their consequences on the Laurentide ice streams, configuration, extent and volume among others. In Section 2, the main features of our model are described; results are shown in Section 3; a discussion is given in Section 4; and the conclusions of this work are presented in Section 5.

## 2   Methods and experimental setup

Numerical experiments are conducted with higher-order three-dimensional ice-sheet model Yelmo (Robinson et al., 2020, 2022). Here, its domain covers the entire LIS topography with a 16 km horizontal resolution. We set 21 unevenly-spaced vertical levels in sigma-coordinates, with higher resolution at the base of the ice sheet. Yelmo uses a higher-order stress approximation known as Depth Integrated Velocity Approximation (DIVA) to compute the horizontal velocity (Goldberg, 2011; Lipscomb et al., 2019). DIVA replaces the horizontal velocity gradients with their vertical averages in the effective strain rate, thus leading to a set of equations similar in accuracy to the Blatter-Pattyn approximation (Blatter, 1995; Pattyn, 2003). The internal ice temperature is determined by the advection-diffusion equation. Anisotropy of the ice is not explicitly modeled so an enhancement factor accounts for crystal orientation on the strain rate (Hooke, 2005; Ma et al., 2010; Pollard and DeConto, 2012; Maris et al., 2014; Albrecht et al., 2020). For simplicity here, the enhancement factor of grounded ice is prescribed to 1.0, whereas floating ice requires a slightly lower value of 0.7 (e.g., Ma et al., 2010).

The total mass balance in Yelmo is governed by three terms: surface mass balance, calving and basal melting. Calving occurs when the ice-front thickness decreases below an imposed threshold (200 m in this study) and the upstream ice flux is not large enough to advect the necessary ice to maintain such thickness (Peyaud et al., 2007). Importantly, basal melting of floating ice is a boundary condition whereas it is calculated internally for grounded ice.

### 2.1   Ice temperature

Yelmo accounts for a classical energy balance governed by an advection-diffusion equation:

$$\frac{\partial T}{\partial t} = \frac{k}{\rho c}\frac{\partial^2 T}{\partial z^2} - u\frac{\partial T}{\partial x} - v\frac{\partial T}{\partial y} - w\frac{\partial T}{\partial z} + \frac{\Phi}{\rho c}, \tag{1}$$

Where $k$ and $c$ are the ice thermal conductivity and specific heat capacity, respectively. The ice temperature evolution is thus determined by vertical diffusion, horizontal and vertical advection, and internal strain-heat dissipation due to shearing $\Phi$:

$$\Phi = 4\nu\dot{\varepsilon}^2, \tag{2}$$

where $\dot{\varepsilon}$ is the effective strain rate and $\nu$ is the ice viscosity.

For grounded ice, when the ice temperature is below the pressure melting point, the prescribed vertical gradient at the base is $\partial T/\partial z = -Q_r/k$, where $Q_r$ is the heat flow at the bedrock surface. The geothermal heat flow $Q_{\text{geo}}$ is then imposed as a boundary condition at 2 km below the bedrock surface. In other words, heat is diffused vertically within the first 2 km of the bedrock, which allows the model to account for the thermal inertia within the bedrock itself (Ritz, 1987).

If the basal ice temperature reaches the pressure-melting point, the temperature is then set to the pressure melting point and the basal mass balance $b_g$ is diagnosed following Cuffey and Paterson (2010):

$$\dot{b}_g = \frac{1}{\rho_i L_i} \left( Q_b + k \left.\frac{\partial T}{\partial z}\right|_b + Q_r \right), \tag{3}$$

where the sign indicates melting when $b_g < 0$. $L_i$ is the latent heat of fusion for ice, $Q_b$ is the basal heat production due to sliding friction and $\partial T/\partial z\big|_b$ is the ice-temperature vertical gradient at the base.

## 2.2 Till hydrology

The subglacial water-flow model assumes a thin film of water. Yelmo then considers a local evolution equation for the basal water content $H_w$ without horizontal advection (considering a hydraulic diffusion coefficient $c_v \sim 10^{-8}\ m^2/s$, e.g., Tulaczyk et al., 2000a). In this case, the non-local term of the time-dependent diffusion equation is assumed to be negligible, yielding the following approximation:

$$\frac{\partial H_w}{\partial t} = \frac{\rho_i}{\rho_w} \dot{b}_g - d_r. \tag{4}$$

Here, $\rho_w$ is the water density, $\dot{b}_g$ is basal mass balance defined in Eq. 3, given by the sum of the frictional heating at the ice-bed interface, and the gradients in heat flow at the base of the ice column and at the bedrock surface (Eq. 4). $d_r$ is the till drainage rate, set to $d_r = 10^{-3}$ m/yr (Bueler and van Pelt, 2015) in the default case which means that its value is generally small compared to $\dot{b}_g$. Negative values of $\dot{b}_g$ are allowed, implying refreezing. The water layer thickness is bounded between zero and a maximum value of $H_{w,\,max}$ (Bueler and Brown, 2009; Bueler and van Pelt, 2015):

$$0 \leq H_w \leq H_{w,\,max}. \tag{5}$$

By default, $H_{w,\,max}$ is set to a constant value of 2 m for simplicity (as in Bueler and van Pelt, 2015).

## 2.3 Friction

Basal shear stress can be generally expressed as a function of the sliding velocity $u_b$ and the effective pressure $N$, i.e., $\tau_b = f(u_b, N)$. The physical properties of the material over which the ice may potentially slide can correspond either to a hard bedrock flow (e.g., Weertman, 1957) or to a Coulomb-plastic rheology. In addition, the influence of the sliding velocity on $\tau_b$ is often represented by a power friction law, although a regularization term $u_0$ accounting for local properties of the bed has been shown to outperform such a power law in several contexts (Joughin et al., 2019; Zoet and Iverson, 2020).

Thus, the basal shear stress (i.e., basal drag) is calculated here via two distinct formulations: a pseudo-plastic power law (Schoof, 2010; Aschwanden et al., 2013) and the regularized-Coulomb formula (Schoof, 2005; Joughin et al., 2019). The former reads:

$$\boldsymbol{\tau}_b = -c_b \left( \frac{|\boldsymbol{u}_b|}{u_0} \right)^q \frac{\boldsymbol{u}_b}{|\boldsymbol{u}_b|}, \tag{6}$$

where $u_0 = 100$ m/yr and $c_b$ is a spatially-variable friction coefficient defined below. We shall focus on two particular cases of the pseudo-plastic law based upon the choice of the exponent $q$. Namely, the linear ($q = 1$; e.g., Quiquet et al., 2018) and the purely plastic law ($q = 0$).

On the other hand, the regularized-Coulomb formula is given by:

$$\boldsymbol{\tau}_b = -c_b \left( \frac{|\boldsymbol{u}_b|}{|\boldsymbol{u}_b| + u_0} \right)^q \frac{\boldsymbol{u}_b}{|\boldsymbol{u}_b|}. \tag{7}$$

Following Zoet and Iverson (2020), we set $q = 1/5$ and $u_0 = 100$ m/yr to ensure a fair transition to the steady-state shear stress supported by the till bed. In the same study the insensitivity of $q$ to the detailed geometry of the bed surface was empirically demonstrated.

The bedrock coefficient $c_b$ is defined as:

$$c_b = \lambda N, \tag{8}$$

where $N$ is the effective pressure (elaborated in Secion 2.4) and $\lambda$ is a function of the bedrock elevation $z_b$ (positive values above sea level):

$$\lambda(z_b) = \begin{cases} 1 & \text{if } z_b \geq 0 \\ \max \left[ \exp \left( -\frac{z_b}{z_0} \right), \lambda_{\min} \right] & \text{if } z_b < 0, \end{cases} \tag{9}$$

where $z_0$ determines the bedrock elevation (positive above sea level) at which $\lambda$ is reduced a factor $1/e$. Additionally, we assume $\lambda_{\min}$ as a lower bound.

Hence, this parametrisation encapsulates the phenomenon by which the occurrence of sliding, as well as its intensity, is favored at low bedrock elevations, in particular within the marine sectors of ice sheets. It is a direct consequence of the presence of soft tills in soils formed mostly by sediments. This is an analogous approach to Albrecht et al. (2020) and Martin et al. (2011), where the bedrock friction is parametrised by a till friction angle set as a function of the bedrock elevation. Notably, this bedrock scaling of $c_b$ (Eq. 9) is a common feature of all approaches presented in Section 2.4, where the same $z_0$

value is employed for every experiment.

## 2.4 Effective pressure

The basal shear stress is not fully determined unless an effective pressure formulation is provided. In this study, two physical scenarios are considered for defining the effective pressure. Namely, in increasing level of complexity: overburden pressure

and a water-dependent effective pressure. The first formulation is a purely mechanical friction approach in which the entire ice weight is considered to compute friction, whereas the second falls within the thermomechanically-coupled friction parametrizations. The latter parametrization is designed to transition from a high friction coefficient (representative of a frozen bed) to a low friction state related to a temperate base. This transition can be solely dependent on the thermal state of the base via potential hydrological processes (i.e., water-dependent approach).

### 2.4.1 Overburden pressure

This is the simplest formulation and merely considers the force exerted by the weight of the overburden ice column on a given point:

$$N = \rho_i g H \doteq P_0. \tag{10}$$

Here, only changes in ice thickness can modify the value of the $N$, increasing with larger ice thicknesses.

### 2.4.2 Water dependent effective pressure

As noted by Brocq et al. (2009), there is a close connection between water depth and sliding speed. This was first acknowledged by Weertman (1964), noting that a water layer with a thickness an order of magnitude smaller than a controlling obstacle size is enough to cause an appreciable increase in the sliding velocity. Tulaczyk et al. (2000a) experimentally demonstrated that the yield strength of till sediments decreases with increasing water content, hence fostering higher velocities. In view of this result, considering the thermal state of the base without the accompanying hydrological processes is a simplification that should be avoided for both soft and hard bedrocks. Several approaches have been considered for simulating the liquid water underneath an ice sheet; here, we we employ the widely used Bueler and van Pelt (2015) effective pressure formulation:

$$\tilde{N} = N_0 \left( \frac{\delta P_0}{N_0} \right)^s 10^{\frac{e_0}{c_t}(1-s)}, \tag{11}$$

where $P_0$ is the overburden pressure, $N_0$ is a constant reference effective pressure, $e_0$ and $c_t$ are empirical constants related to till properties, $s = H_\mathrm{w}/H_\mathrm{w,\,max}$ is the till saturation and $\delta$ is the minimum overburden pressure fraction for a completely saturated till. Following Bueler and van Pelt (2015), we choose a value of $\delta = 0.02$, so that a fully saturated till yields an effective pressure equal to $2\%$ of the overburden pressure exerted by the ice.

In reality, the effective pressure $N$ cannot exceed the overburden pressure $P_0$ for any sustained period, shaping $P_0$ into an upper limit:

$$N = \min\left\{ P_0, \tilde{N} \right\}. \tag{12}$$

Therefore, the effective pressure of the till is an exponential transition between these two extreme cases: the entire weight of the ice column $N = P_0$ for a fully drained till $s = 0$ and a minimum value $N = \delta P_0$ for saturated conditions $s = 1$.

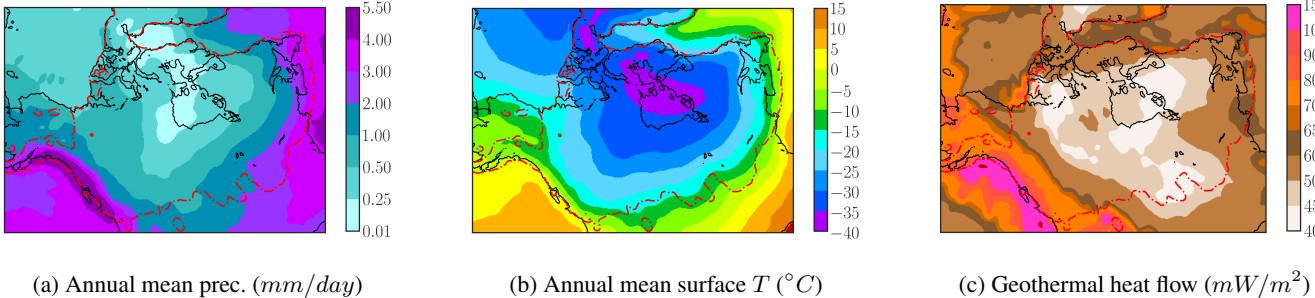

(a) Annual mean prec. $(mm/day)$    (b) Annual mean surface $T$ $(^\circ C)$    (c) Geothermal heat flow $(mW/m^2)$

**Figure 1.** Mean imposed climate fields. LGM constant conditions define the external climatic forcing so that none of these boundary conditions exhibit temporal dependency. Red dashed line shows maximum reconstructed LIS extent (ICE-6G).

## 2.5 Experimental setup

In order to investigate the effect of different friction formulations on the simulation of the LIS at the LGM, two sets of experiments were carried out. First, the effective pressure $N$ is assumed to solely depend on the overburden pressure (Section 2.4)
exerted by the ice column. In this simple scenario (purely mechanical friction), we consider three different basal friction laws with different dependencies of the basal shear stress on the sliding velocity: linear, power law (purely plastic) and regularized-Coulomb parametrizations. Second, for the most comprehensive basal friction parametrization law (i.e., regularized-Coulomb), we allow for thermomechanical coupling of the sliding by introducing an additional dependency of $N$ on the thermal state of the base via the water-dependent formulation.

Constant LGM conditions define the climatic boundary conditions. To this end, atmospheric temperature and precipitation are climatologies obtained from the mean of the output of the 11 General Circulation Models (GCMs) participating in the Paleoclimate Modelling Intercomparison Project Phase III (PMIP3) as part of the Coupled Model Intercomparison Project Phase 5 (CMIP5; Taylor et al., 2012) (Fig. 1a and 1b). The geothermal heat flow is also a spatially-variable boundary condition in our simulations and it is acquired from Shapiro and Ritzwoller (2004) (Fig. 1c).

Additionally, the initial bedrock elevation is taken from the RTopo2.0.1 present-day Earth topography dataset (Schaffer et al., 2016). The bedrock topography evolves under glacial isostatic adjustment (GIA) via the elastic lithosphere-relaxed asthenosphere (ELRA) method (Meur and Huybrechts, 1996) with a spatially-constant relaxation time of 3000 years.

**Table 1.** Parameter choice employed in our simulations and sample ranged. The friction exponent $q$ is taken from Zoet and Iverson (2020) for the regularized-Coulomb case.

|  | Linear | Plastic | Coulomb | Explored range |
|---|---|---|---|---|
| $q$ | 1 | 0 | 1/5 | N/A |
| $z_0$ $(m)$ | $-100$ | $-100$ | $-100$ | $[-800, 200]$ |
| $u_0$ $(m/yr)$ | 100 | 100 | 100 | $[25, 250]$ |

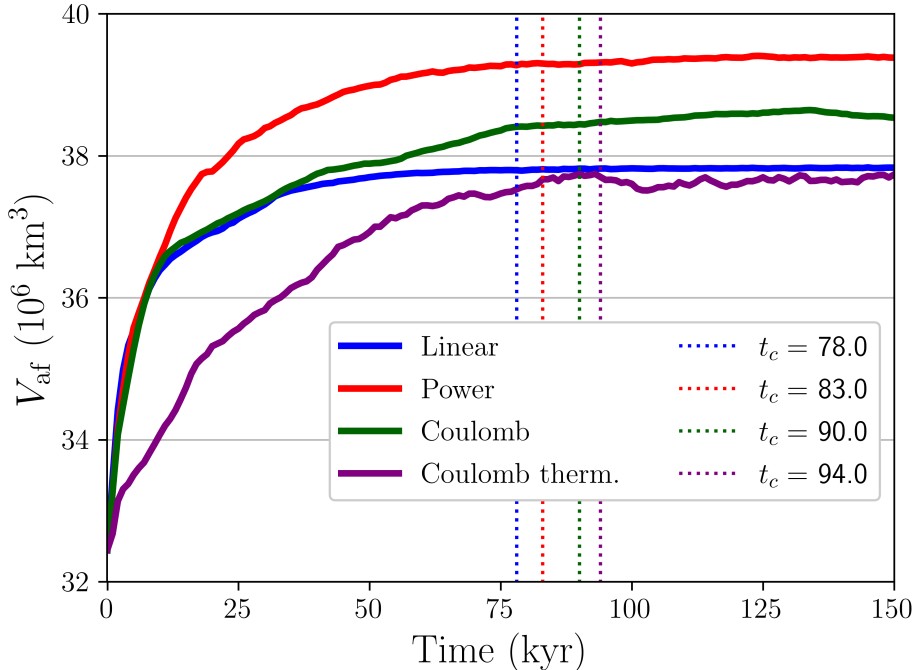

**Figure 2.** Ice volume above flotation $V_{\mathrm{af}}$ for the main simulations. Vertical dashed lines represent the changepoint (i.e., the transition from transitory to stationary regime) for each time series as determined by the two-phase linear regression (details in Appendix A). For $t > t_c$, a constant equilibrium volume is reached in all cases.

Finally, we sampled a broad parameter range of $z_0$ values and then tuned so as to obtain an ice stream network that resembles previous mapping inventories (e.g., Fig. 2 in Margold et al., 2015). Hence, we first defined an ice stream as a set of grid points that satisfy $u_{\mathrm{b}}/u_{\mathrm{def}} > 10$. In other words, ice streams are here defined as regions of the ice sheet where the sliding contribution is, at least, one order of magnitude greater than ice deformation. It must be stressed that no particular LIS volume value was targeted but rather, the model is tuned based on the dynamics. The same $z_0$ value is then employed throughout the study (see Table 1). This approach provides good qualitative results and facilitates comparison among the model formulations used here.

Simulations throughout this study ran for 200 kyr to ensure a smooth equilibration from the initial state. An initial ice thickness of 1000 m is imposed over bedrock above sea level in North America above $50°$N to urge the spin up. The necessary length of the spin-up is quantified by a two-phase linear regression (Hinkley, 1969, 1971), i.e. a statistical test for detecting a change in behaviour of a variable time series (i.e., the so-called *changepoint*, details in Appendix A). Namely, we applied the two-phase regression model to the ice-sheet volume above flotation time series so as to determine the equilibration time (Fig. 2). The average equilibration time of all simulations herein presented reads $\bar{t}_{\mathrm{eq}} = 86.3$ kyr.

Thus, the first 100 kyr were assumed to represent model spin-up and are not considered in the analysis here. The remaining 50 kyr are shown in the figures below. All simulations were performed with a horizontal grid resolution of 16 km.

## 3    Results

Two main experiments were performed throughout this study accounting for each effective pressure formulation: purely mechanical friction (overburden) and thermomechanically coupled (i.e., a water-dependent parametrization), as described above. Each of this cases is described in the following sections.

In general, our simulations largely agree in extent with prior reconstructions (Stokes et al., 2016; Stokes, 2017). This result is not expected *a priori* since we tuned Yelmo ice-sheet model to obtain a fully-developed ice stream network (e.g., Margold et al., 2014, 2015) rather than to match a certain volume and extent estimation (Section 2.5). It is worth noting that Margold et al. (2015) already stressed that no inferences on the timing of ice stream operation are possible because a small number of the mapped ice streams have any chronological control. Yet, it is clear that their mapped ice stream tracks represent a time-transgressive imprint of evolving ice stream trajectories, i.e. they can not have all operated at once. Nonetheless, some broad spatial patterns appear and we further exploit this fact to compare our simulations. Potential timing inconsistencies are thus inevitable, though the time-transgressive inventory remains as an appropriate reference for the simulated ice streams.

Further comparison with Margold et al. (2014) ice stream inventories was performed by re-projecting their data to the same coordinate system used in Yelmo LIS simulations. Namely, from a Lambert conformal conic projection (EPSG:3978) to polar stereographic.

As we shall note, the particular basal friction dependency on the sliding velocity leaves the ice extent and total volume nearly unchanged even though it strongly influences the ice stream configuration. On the contrary, the thermodynamical treatment of the ice-sheet base entails significant differences mainly in total volume.

### 3.1    Purely mechanical friction

We will first describe the reconstruction of our simulated LIS under LGM conditions for the three basal friction laws (linear, plastic and regularized-Coulomb) and no thermal coupling of the basal sliding. All simulations are numerically stable and reach constant equilibrium values within the first 100 kyr. Figure 3 shows important differences in the dynamic configuration of the ice sheet among the three cases.

In the linear case, ice streams appear to be widely distributed, far beyond the expected locations from prior reconstructions (e.g., Margold et al., 2015), thus differing from the purely plastic and regularized-Coulomb scenarios (Fig. 3). As a result, horizontal velocities are generally high, even far from topographic troughs, allowing for strong lateral ice advection and both the ice thickness and the volume above flotation reach a minimum (Table 2). Rapid sliding also occurs near the margins where the continuity equations favours ice advection partially due to a large calving term. A more comprehensive dependency of the basal stress on the sliding velocity (e.g., a plastic or a regularized-Coulomb) shows that a fully-developed ice-stream network can be simulated even for a simple overburden formulation (Fig. 3e, 3f). Unlike the linear case, ice streams in the latter case are constrained spatially to lower troughs as a result of friction saturation at higher velocities (Joughin et al., 2019), allowing fast streams to develop mainly where soft sediments are assumed to enhance sliding (Eq. 9).

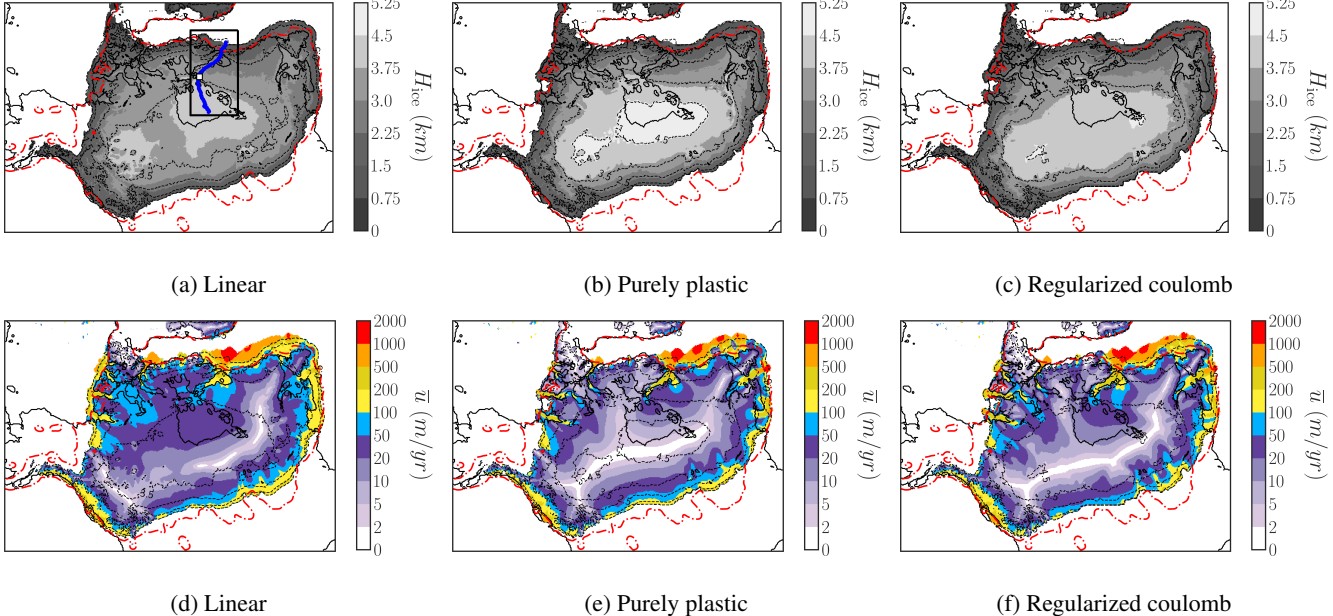

**Figure 3.** First row, LIS ice thickness in kilometres; second, vertically averaged horizontal velocity. Each column corresponds to one friction law, from left to right: linear, purely plastic and regularized-Coulomb. Red dashed line shows maximum reconstructed LIS extent (ICE-6G). Black dashed line shows ice thickness contours in kilometres at values of 1.0, 2.5, 3.0, 3.5, 4.0 and 4.5 km. In panel (a), the black rectangle defines the Hudson Strait subdomain as referred to in the text. A blue solid line represents the Hudson ice stream section and a black solid contour denotes the present day coastline. Time series evaluated over a 9-grid-point square are centred in the white dot.

In terms of the ice-thickness dome configuration, all reconstructions show a multi-domed configuration with two rela-
tive maxima: the eastern dome, centred over Hudson Bay and the western dome, over Lake Claire. Nevertheless, the mini-
mum/maximum thicknesses are found for the linear and the power law scenarios respectively, whilst leaving the regularized-
Coulomb case as an intermediate reconstruction. This is presumably caused by a further inland penetration of the Northwest
ice streams in the regularized-Coulomb scenario compared to the purely plastic case. For the linear friction, we find generally
higher velocities in the northwest and inner LIS. This translates into a larger amount of ice advected, consequently reducing
the ice equilibrium thickness (mass balance equation).

The basal friction law has implications for the thermal state of the base even in the absence of thermomechanical coupling
(Fig. 4). The LIS appears to be mostly temperate, except for the south-eastern region of the Canadian Shield. The spatial
distribution of the basal temperature can be understood given that the ice sheet behaves as a thermal insulator. The nearly
fully temperate base in the power law corresponds to the thickest LIS reconstruction. For the base to remain frozen two main
requirements must be met: low sliding velocities (i.e., low frictional heat) and low geothermal heat flow (Fig. 1c). The former
is demonstrated in Fig. 5 for all three cases, whereas a strong correlation between frozen basal regions of the LIS and minimum
geothermal heat flow values (Shapiro and Ritzwoller, 2004) supports the latter.

Figure 6 shows that the dynamic state of the ice sheet is highly sensitive to the particular function $\tau_b(u_b)$. We notice that the regularized-Coulomb case appears to be an intermediate scenario between the linear and the purely plastic. However, there is a distinct common feature of the Coulomb and purely plastic cases: a linearly increasing lower boundary of $\tau_b$ for velocities $u_b > 200$ m/yr. This can be explained by the minimum value of the friction coefficient (to avoid spurious velocities). This value is a constant so that the basal shear stress becomes proportional to the sliding velocity, thus giving rise to a linear dependency. The behaviour is only visible for high velocities given the nature of minimum shear stress.

From an energy balance perspective, the dissipated frictional heat $Q$ provides an idea of how the mechanical energy is distributed in the system (7). Our simulations have attained a steady state so all the energy that enters our system must be dissipated. The ice mass moves as a consequence of its own weight, i.e. the potential energy transfers to kinetic energy via the surface elevation slope (driving stress). The equilibrium velocity field is then maintained by the new ice accumulated on the domain. In the linear case, most of the kinetic energy is dissipated by thin ice with relatively large shear stresses. The purely plastic scenario yields a more distributed energy dissipation, where thick ice ($H > 3.0$ km) also has a significant contribution. As mentioned before, the Coulomb case appears as an intermediate physical description, thin ice dissipates more heat compared to the purely plastic scenario, yet large thicknesses have a significant frictional heat unlike in the linear case.

The basal stress distribution for different ice thicknesses (Fig. 6) may seem counterintuitive given that, for a fixed velocity, lower $\tau_b$ values are generally reached for thicker grid points. Yet this can be understood in terms of the bedrock characteristics (Eq. 8) as follows. Thick ice within the LIS is unable to reach high velocities unless it is restricted to low elevations (as $c_b$ approaches its minimum). On the contrary, if we consider low thicknesses, the same velocities can be found for considerably higher $c_b$ values (since $N = \rho g H$ is smaller). In other words, for a particular velocity, thinner ice yield higher basal stress due to the bedrock characteristics.

The different ice-sheet dynamics result in different configurations for the LIS (Table 2). In general, our simulations are consistent with our current knowledge of the LIS during the LGM, yet it is worth noting certain aspects of each parametrization. The fact that the linear law leads to the lowest values of ice volume (above flotation) and ice thickness can be explained by recalling Joughin et al. (2019). For low velocities (i.e., the centre of the LIS), the linear friction law (Fig. 6a) yields lower $\tau_b$ values than a plastic/Coulomb law (Fig. 6b and 6c). Such inland points consequently have higher velocities, thus advecting ice towards the margins and decreasing the equilibrium ice thickness. This entails a straightforward reduction in the effective pressure $N$. As a result, the basal friction coefficient reaches a minimum. In contrast, only minor differences in ice volume are found between the more comprehensive plastic law and regularized-Coulomb parametrizations.

Lastly, we present longitudinal sections of the Hudson Strait ice stream for the linear, the purely plastic and the regularized-Coulomb friction laws (Fig. 8). The location of the points of the section was selected on the basis of a maximum velocity criterion so that the section lies in the centre of the ice stream and extends from Hudson Bay to the grounding line (Fig. 3a). As we would expect, results with a linear friction law differ most. Particularly, deformation velocities close to the margin are the highest among the three laws herein considered as a result of an absent upper bound in the basal shear stress. Basal velocities near the dome of the LIS are also higher for a linear case given that $\tau_b(u_b)$ approaches zero more rapidly for $q = 1$ than for $q < 1$ (Eq. 6). Therefore, the ice thickness is a minimum as dictated by the continuity equation (consistent with Table 2). A

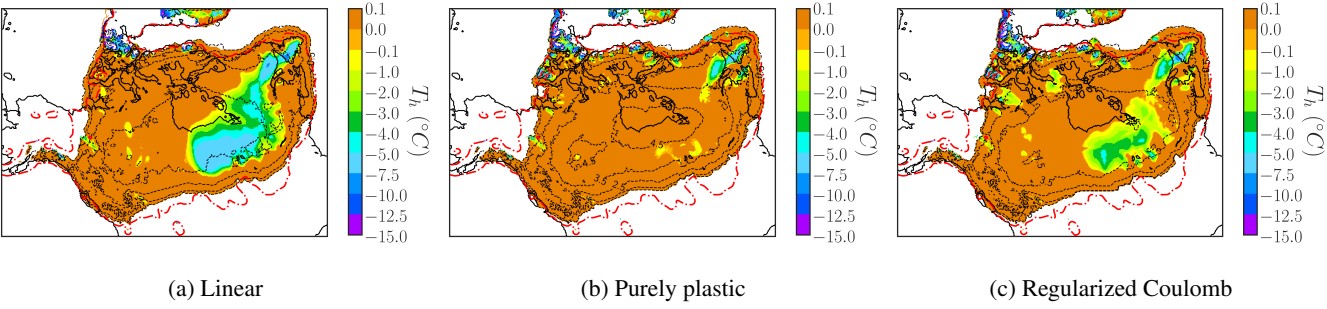

(a) Linear          (b) Purely plastic          (c) Regularized Coulomb

**Figure 4.** Homologous ice-sheet base temperature (ºC) for the three friction laws: (a) linear, (b) purely plastic and (c) regularized-Coulomb.

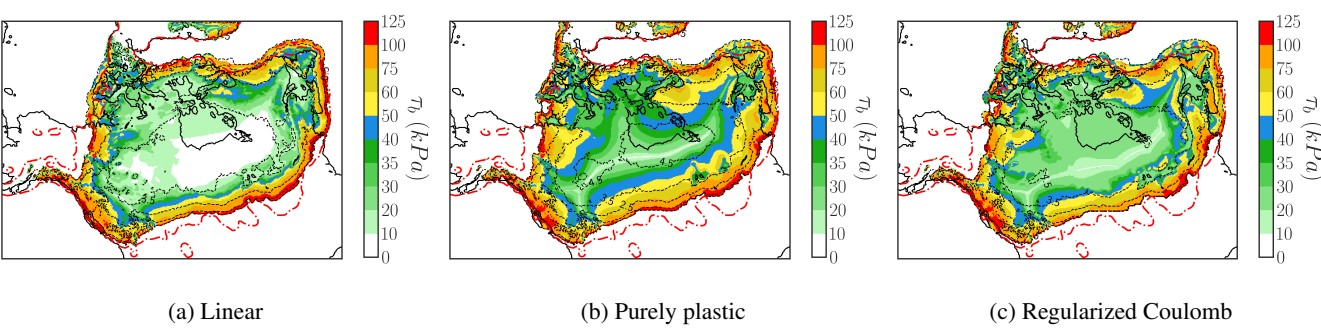

(a) Linear          (b) Purely plastic          (c) Regularized Coulomb

**Figure 5.** LIS shear stress $\tau_b$ (Pa) for the three friction laws: (a) linear, (b) purely plastic and (c) Regularized-Coulomb. Red dashed line shows maximum reconstructed LIS extent (ICE-6G). Black dashed line shows ice thickness contours in kilometres.

subtle difference between the power law and the regularized-Coulomb case is visible on the surface elevation slope. In general, and particularly near the dome, the slope is slightly steeper in the power law case and the consequences are noticed in a higher

deformation velocity (dashed blue line) in Fig. 8b than 8c.

**Table 2.** Ice volume above flotation $V$, extent $A$, maximum ice thickness $H_{\max}$, spatially averaged basal temperature $\overline{T}_{\mathrm{b}}$ and sliding velocity $\overline{u}_{\mathrm{b}}$ for the three friction parametrizations under consideration. Average quantities carry between brackets the corresponding standard deviation value.

| Therm-coupled friction | Basal friction law | $\mathbf{V}$ ($10^6$ km$^3$) | $\mathbf{A}$ ($10^6$ km$^2$) | $H_{\max}$ (km) | $\overline{T}_{\mathrm{b}}(\sigma)$ (ºC) | $\overline{u}_{\mathrm{b}}(\sigma)$ (m/yr) |
|---|---|---|---|---|---|---|
| | Linear | 36.9 | 16.5 | 4.1 | $-1.0$ (2.3) | 33.8 (114.6) |
| No (overburden) | Purely plastic | 39.5 | 19.5 | 5.0 | $-0.7$ (1.7) | 24.4 (137.2) |
| | Regularized-Coulomb | 38.1 | 16.3 | 4.6 | $-0.8$ (1.8) | 28.4 (127.8) |
| Yes (water dependent) | Regularized-Coulomb | 33.5 | 16.0 | 4.3 | $-0.7$ (1.6) | 27.7 (139.0) |

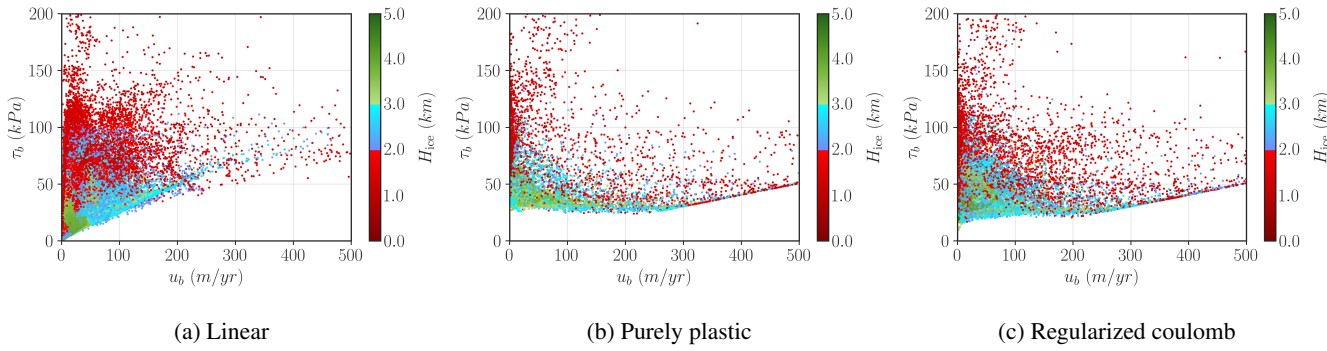

**Figure 6.** Scatter plot of $\tau_b(u_b)$ phase space for three different basal friction laws: (a) linear, (b) purely plastic and (c) regularized-Coulomb. Every dot represents a pair $(u_b, \tau_b)$ evaluated in a single grid point.

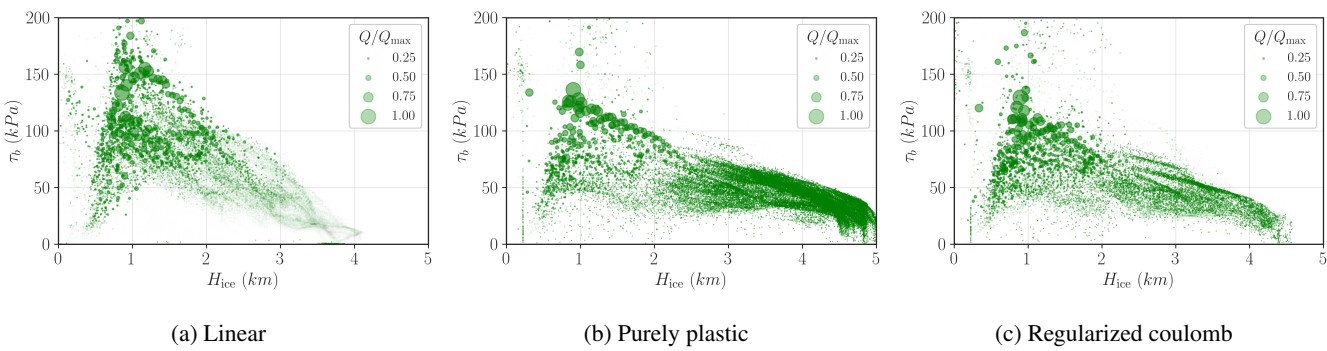

**Figure 7.** Frictional heat distribution as a scatter plot of $\tau_b(H_{\mathrm{ice}})$ for three different basal friction laws: (a) linear, (b) purely plastic and (c) regularized-Coulomb. Every dot represents a pair $(H_{\mathrm{ice}}, \tau_b)$ evaluated in a single grid point. The marker size represents the normalised frictional heat $Q/Q_{\mathrm{max}}$, where $Q = u_b \tau_b$ and $Q_{\mathrm{max}}$ is the maximum value of each simulation.

## 3.2 Thermomechanically coupled friction

Next we investigate the effect of coupling basal friction to the thermal state of the base by comparing the simulated LIS under LGM conditions for the water-dependent parametrization with the purely mechanical friction formulation. A regularized Coulomb friction law is employed throughout this section. In terms of ice thickness, there is no clear distinction between a
purely mechanical friction approach (Fig. 3f) and the thermomechanically coupled case (Fig 10) besides a minor decrease. More precisely, Table 2 shows slight differences in total ice volume and extent: the thermomechanically coupled simulations show a smaller extent and therefore a lower volume given that the ice thickness remains nearly identical. Nevertheless, such decrease brings our simulation closer to previous reconstructions (Fig. 9). Yet the ice extent remains in the upper limit compared to prior studies. This further suggests that, for our particular parameter choice, a thermomecanically-coupled fricton may be
necessary to obtain a realistic LIS extent.

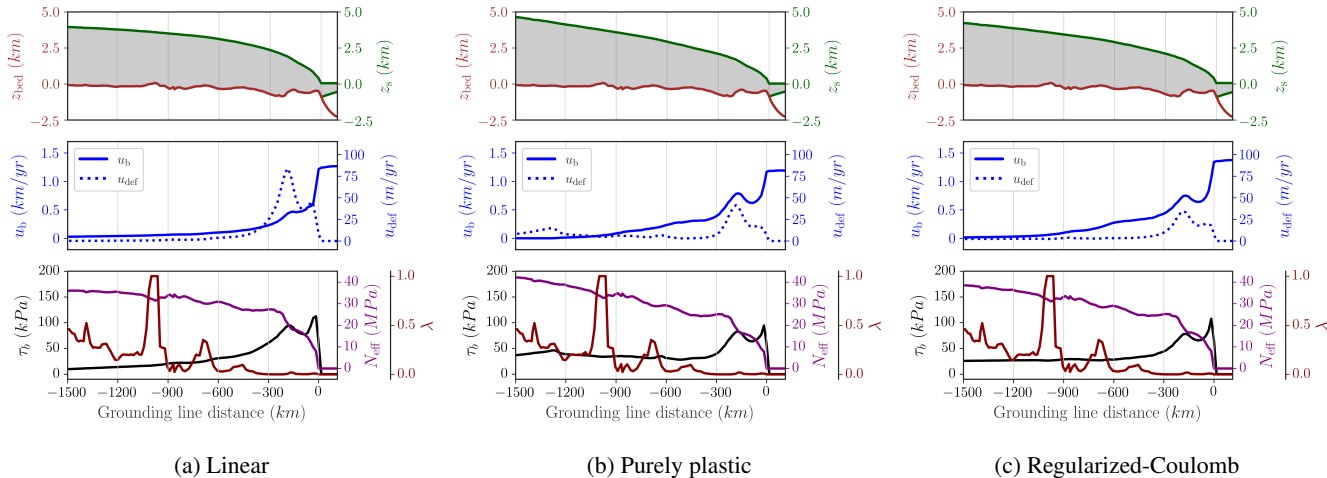

**Figure 8.** Section along Hudson Strait ice stream (as noted in Fig. 3a) for purely mechanical basal frictions: linear, purely plastic and regularized-Coulomb. Green, LIS surface elevation; brown, bedrock height; blue, horizontal velocity (sliding and deformation contributions); purple, effective pressure and black, basal shear stress.

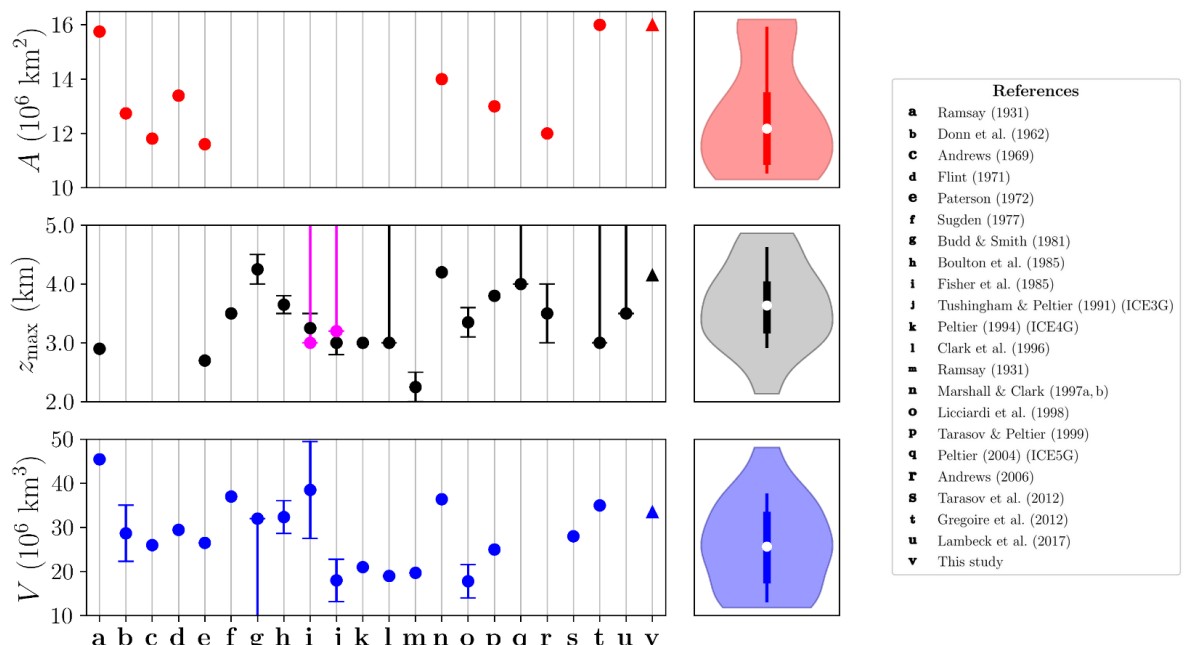

**Figure 9.** Comparison of reconstructed LIS ice extent, maximum elevation and volume respectively. The current work estimations are given by triangle markers. Magenta dots show maximum ice-sheet elevation for the soft bed models.

It is illustrative to build a streaming mask to perform a qualitative comparison among parametrizations as well as previous inventories (e.g., Margold et al., 2015). We therefore define sliding regions as those points that satisfy the condition $u_\mathrm{b}/u_\mathrm{def} >$ 10, thus ensuring that ice flow due to deformation is, at least, one order of magnitude lower than the sliding contribution. In terms of this streaming mask (Fig. 10b), we generally simulate the most significant ice streams present in recent mapping inventories and comprehensive reviews of the LIS (e.g., Margold et al., 2014, 2015).

The thermomechanically coupled friction formulation entails fundamental changes in the LIS configuration and thermal state of the base. A direct inspection of Fig. 3f as compared to Fig. 10a further shows the implications in the simulated ice stream configuration and notable improvement is found in the Hudson Strait ice stream and tributary.

The probability density functions $P(u_\mathrm{b})$ and $P(T_\mathrm{b})$ (Fig. 11) further explore the differences among friction law formulations both for an overburden and a water-dependent effective pressure. For the linear law, we find the coldest ice base on average (see Table 2) as the tail of the distribution reaches leftmost values compared to a power or Coulomb formulation. Notably, these two last friction laws show minor differences in terms of $P(u_\mathrm{b})$ and $P(T_\mathrm{b})$, showing physically equivalent reconstructions in terms of probability densities. On the contrary, when the basal friction is coupled with thermodynamics via Eq. 11, we note a shift towards higher velocities $P(u_\mathrm{b})$ for low velocities (i.e., $u_\mathrm{b} < 250$ m/yr), thus implying a speed-up of the slower regions of the ice sheet. Consequently, the outflow of ice becomes larger and the equilibrium thickness is reduced compared to the Coulomb overburden scenario (Table 2).

When the basal friction is thermomechanically coupled (Table 2), the LIS extent is reduced and the maximum ice thickness is lower, leading to a smaller equilibrium volume. This is explained through the decrease in basal friction. In this case, there is an additional degree of freedom that may yield a reduction in basal friction: the effective pressure. All temperate grid points undergo a reduction in their effective pressure (and consequently in the basal stress) by up to a $10\%$ of their original value. As a result, the stress balance will yield higher velocities and a lower equilibrium thickness for a fixed set of boundary conditions. On the contrary, in the purely mechanical friction case, the value of $c_b$ is determined solely by the bedrock elevation, which does not change significantly over the course of the experiment.

Nevertheless, the equilibrium volume, relevant for the sea level contribution, does not encapsulates all the relevant information about the LIS, especially for the Hudson subdomain. Notably, the ice volume in the Hudson subdomain (as defined by the black rectangle in Fig. 3d) reaches a constant equilibrium value both in the purely mechanical and thermomechanically coupled experiments. Likewise, the vertically averaged horizontal velocity also attains a constant value, yet slightly higher due to the water-dependent effective pressure for the aforementioned mechanism.

Global variables such as the total LIS volume are not the only ones that undergo changes when the basal friction is further coupled to thermodynamics. This result is captured by Fig. 12a. Unlike its counterpart in the purely mechanical case (Fig. 6c), we find an interesting behaviour of the non-monotonic minimum shear stress values in the low velocity regime ($u_b < 150$ m/yr) when the basal friction is coupled with thermodynamics. Nonetheless, all points taking part in this minimum shear stress region correspond to a fully drained till. Hence, explicit water changes do not explain the difference in behaviour. Presumably, we argue that those points with lowest $\tau_b$ cannot be reached given the new stress balance (i.e., the SSA equations) is changed if we account for $N_\mathrm{eff}$. Since the SSA solution is non-local, the particular shape of $\tau_b(u_b)$ can be modified by a water-dependent

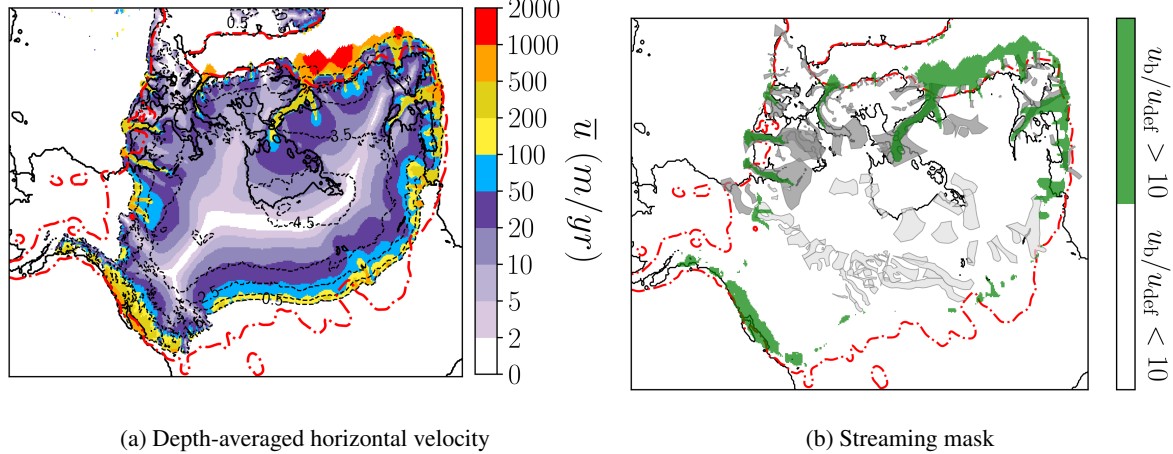

(a) Depth-averaged horizontal velocity

(b) Streaming mask

**Figure 10.** Left panel, LIS depth-averaged horizontal velocity; right panel, spatial mask (green) depicting the two ice flow regimes overlaid with Margold et al. (2014) ice-stream inventory (polygons). Solid polygons correspond to land terminating (light grey) and marine terminating (dark grey) ice streams respectively. Streaming grid points meet the condition $u_{\mathrm{b}}/u_{\mathrm{def}} > 10$ so that the flow due to ice deformation represents, at highest, a contribution one order of magnitude below sliding. Both fields are shown for a water-dependent effective pressure. Red dashed line shows maximum reconstructed LIS extent (ICE-6G). Black dashed line shows ice thickness contours in kilometres of 1.0, 2.5, 3.0, 3.5, 4.0 and 4.5 km.

effective pressure even for regions that are fully drained. This implicit effect would be a direct consequence of the non-local nature of the SSA solutions in regions where the water content remain constant.

It is also illustrative to compare the Coulomb friction law for both a purely mechanical friction and the thermomechanically coupled case from a frictional heat perspective (Fig. 7c and 12c, respectively). When the basal friction is then coupled with the thermal state of the base via a the water layer thickness $H_{\mathrm{w}}$, we notice two main changes. First, the shear stress values are generally reduced and the the thicker regions of the LIS contribute more to frictional heat dissipation (larger region covered in green for $H > 3.0$ km).

It is clear from Fig. 12b that, for an effective pressure that depends on basal water thickness, sliding occurs when the till is saturated in water. This requires a sustained supply of heat (e.g., basal frictional heat, geothermal heat flow, etc.) to melt enough water so as to keep a saturated till, thus surpassing the drainage rate and eluding refreezing (due to heat diffusion-advection, Eq. 2). This is unlikely to occur in the central region of the ice sheet where neither low troughs nor high surface slopes are present, consequently yielding low driving stresses and basal frictional heat.

## 4 Discussion

In general, the ice sheets simulated herein are consistent with our knowledge of the LGM Laurentide ice-sheet state. Qualitatively, this can be seen by a comparison of Fig. 10b with previous reconstructions of LIS ice dynamics (e.g., Margold et al.,

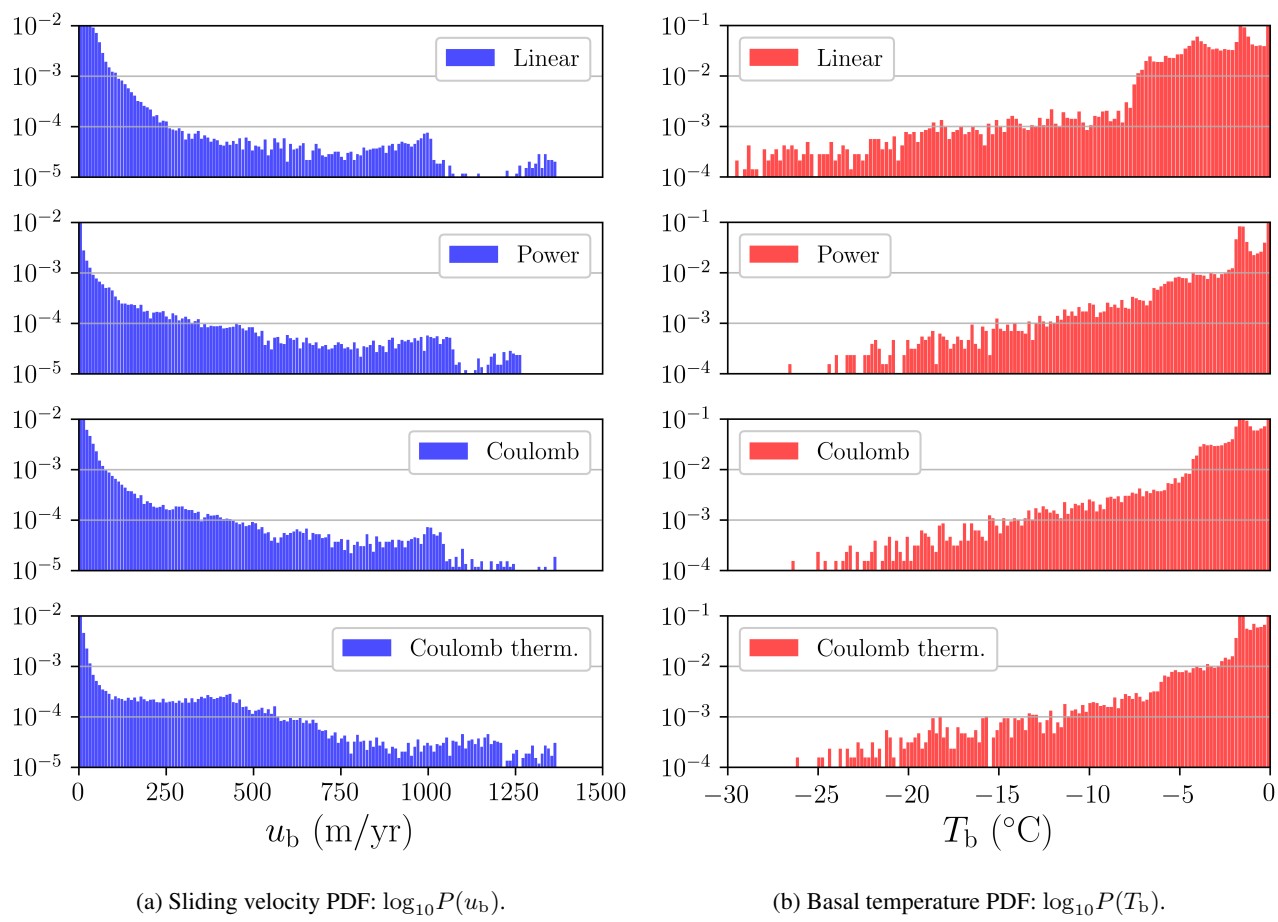

(a) Sliding velocity PDF: $\log_{10}P(u_b)$.

(b) Basal temperature PDF: $\log_{10}P(T_b)$.

**Figure 11.** Probability density functions (PDF). Each row represent a different friction formulation. From top to bottom: linear, power law, regularized-Coulomb and regularized-Coulomb with a water-dependent effective pressure formulation. Note the difference in y-axis limits.

2015; Stokes et al., 2016). Notably, the main ice streams of the LIS (i.e., Amudsen Gulf, M'Clure Strait, Massey Sound, Gulf of Boothia, Lancaster Sound and Hudson Strait) are present in our simulation even in the absence of thermomechanical coupling (Fig. 3e and 3f). However, both the configuration of ice streams and the total ice sheet volume are found to be strongly dependent on the basal friction formulation.

In particular, the linear basal friction law clearly yields significantly lower shear stress values compared to the other formulations (Fig. 3). Despite the fact that both ice extent and volume do not fall far from previous studies, relatively high velocities are found further inland in ice streams along the northern LIS and are not fully constrained to lower troughs (Fig. 3d). As a result, the ice sheet under this parametrization exhibits a minimum volume and a simple-domed ice sheet that resembles past reconstructions that ignore deformable beds (e.g., Denton, 1981). This can be understood as follows. The equilibrium thickness 355  is in fact explicitly dependent on the horizontal velocity via the continuity equation, thus reaching a minimum value when the

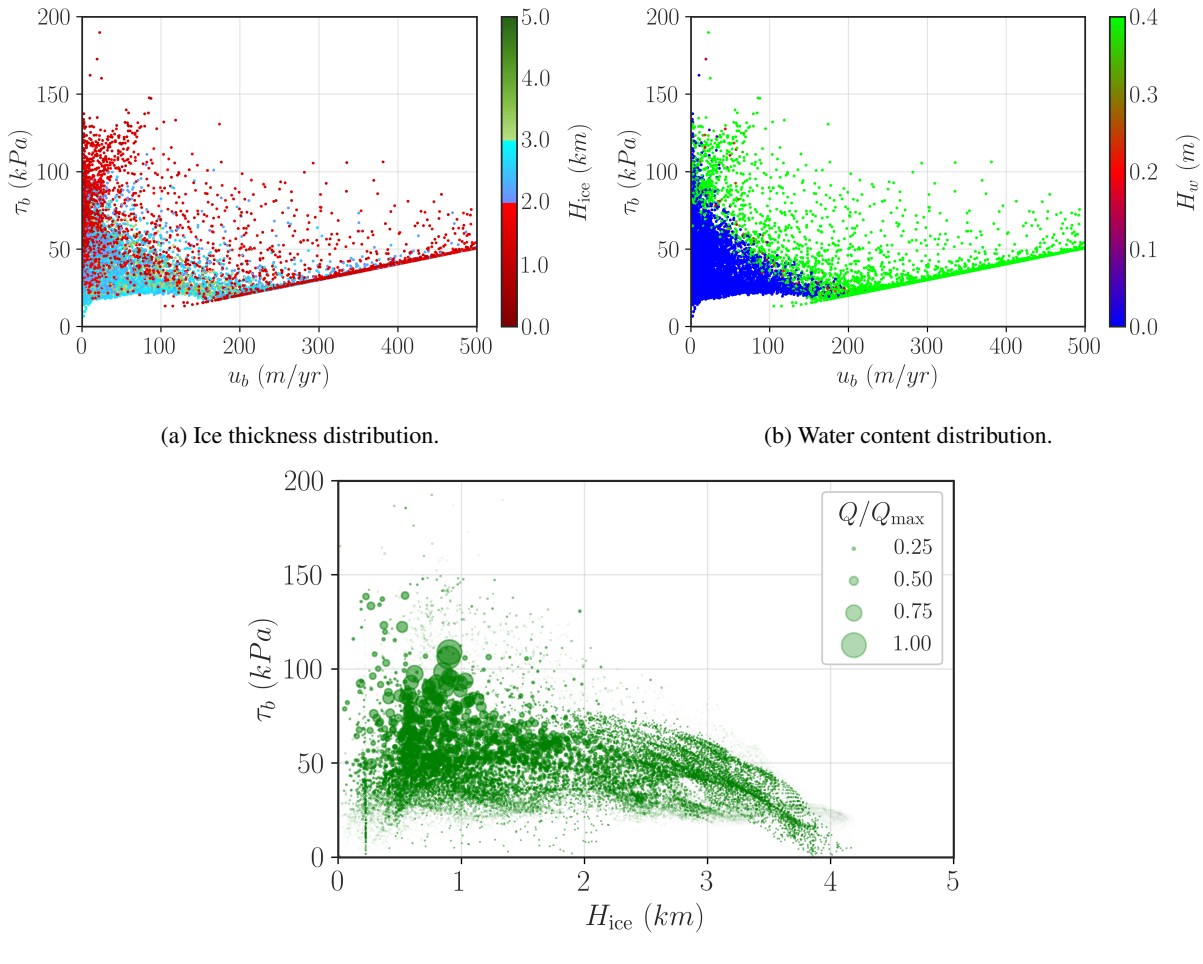

(a) Ice thickness distribution.

(b) Water content distribution.

(c) Frictional heat distribution.

**Figure 12.** Scatter plot of $\tau_b(u_b)$ phase space for the water-dependent effective pressure formulation and coloured according to (a) ice thickness and (b) basal water content. Every dot represents a pair $(u_b, \tau_b)$ evaluated in a single grid point. Panel (c) shows a scatter plot of $\tau_b(H_{ice})$ for the water-dependent effective pressure, where each dot represents a pair $(H, \tau_b)$ evaluated in a single grid point. The marker size depicts the normalised frictional heat $Q/Q_{max}$, where $Q_{max}$ is the maximum frictional heat value.

velocity is high for a fixed set of boundary conditions (i.e., the accumulation rate). Hence, the maximum ice thickness yields its lowest value in this reconstruction.

These results could lead to the hypothesis that rapid ice-streaming spatially constrained to lower troughs requires a thermal coupling with the base. Nevertheless, the absence of a thermomechanical coupling solely exhibits a fully-developed and spatially constrained ice stream structure when a more realistic function for $\tau_b(u_b)$ is provided (i.e., a power-law or a regularized-Coulomb). Although the same ice extent appears to be reached independent of such a function, it closely matches the ICE-6G reconstruction. Thus, thermomechanical coupling is not necessary to simulate a fully-developed ice-stream network in the expected locations. In fact, a more realistic $\tau_b(u_b)$ is sufficient to find rapid streaming regions spatially constrained to low troughs as is the case for a purely plastic or a regularized-Coulomb parametrization. Significantly lower basal friction values are yielded by the former, yet the dynamic configuration of the ice sheet seems almost identical. Despite these similarities, from a purely thermodynamic perspective of the ice-sheet base, the choice of $\tau_b(u_b)$ is fundamental even when thermal coupling is not considered. This is presumably due to the insulator effect of a thicker ice sheet from the colder atmosphere (see maximum equilibrium thickness in Table 2).

Fundamental changes are noticed when the basal friction parametrization is coupled with the thermal state of the base (Fig. 11 and 12). Rapidly-flowing ice streams are present in expected locations, such as through Hudson Strait, Amundsen Gulf, M'Clure Strait, Lancaster Sound and Gulf of St Lawrence (Margold et al., 2015). Consequently, both the total volume and the equilibrium ice thickness are reduced. Overall, the simulated ice sheet closely matches the reconstructed ICE-6G extent, even though it is somewhat lower than for the overburden case. All friction laws herein presented yield a multi-domed ice sheet where two independent domes are found (western and eastern) irrespective of the thermomechanical coupling. The total ice volume, in terms of contribution above flotation, is $33.5 \times 10^6$ km$^3$ (Table 2). This value is larger than the estimate given by Sims et al. (2019) ($30.4 \pm 2.7 \times 10^6$ km$^3$), though close to Gregoire et al. (2012) ($35 \times 10^6$ km$^3$). Furthermore, no large volume changes are found either in the entire LIS nor the Hudson region that would resemble binge-purge oscillations (MacAyeal, 1993a).

Not only does the Bueler and van Pelt (2015) effective pressure formulation couple ice dynamics with the thermomechanical state of the base, but also the amount of liquid water is considered to compute the effective pressure. Figure 6 shows a significant difference in terms of the horizontal velocity and the basal friction coefficient. As described above, the simulated ice sheet also appears to be a multi-domed configuration with two relative maxima that resemble the previous result (western and eastern domes). Even so, the ice-stream structure strongly differs from the purely mechanical friction approach. First, the ice streams are more restricted spatially, in the sense that they do not propagate as far inland. Second, even for non-streaming regions, $\tau_b$ values are generally higher for the water-dependent effective pressure formulation.

The fact that all our reconstructions share a multi-domed equilibrium configuration resembles the prevailing approach of LIS reconstructions that have accounted for lubricated basal conditions, in which the ice sheet over Hudson Bay was consequently thinner and less steeply sloped (e.g., Boulton et al., 1985; Fisher et al., 1985). Nonetheless, the surface elevation over Hudson Bay was substantially lower in those cases, with a maximum elevation above present sea level of 3.0-3.5 km, in contrast to our $\sim 4$-5 km thickness. This comparison must be taken with caution since surface elevation and ice thickness do not represent the

same magnitude. Yet, it is possible to have an approximate comparison among reconstructions by also looking at the volume differences. Boulton et al. (1985) spans a volume of $33\text{-}44\times10^6$ km$^3$, substantially larger than the $21.1\text{-}25.9\times10^6$ km$^3$ range of Fisher et al. (1985) for the hard bed model in both cases (Fig. 9). Our particular volume values fall within Boulton et al. (1985)'s range. In terms of volume and ice extent, results from the water-dependent effective pressure formulation yield a slightly larger

ice volume as a result of narrower and shorter ice streams that consequently advect less ice towards the edges. This dynamic distinction is significant for ice extent given that the reconstructions exhibits the lowest ice extent value ($16.0\times10^6$ km$^2$).

Notably, the most realistic parametrization (a water-dependent effective pressure formulation) shows an interesting behaviour that deviates from the cases using the overburden pressure approach. For low velocities, the shape of $\tau_b(u_b)$ is almost identical to the overburden case. Nevertheless, for higher velocities ($u_b > 80$ m/yr), the phase space $\tau_b(u_b)$ differs from the purely

mechanical reconstructions, where quite low basal stresses are yielded. Figures 12a and 12b then establish the distribution of ice thickness and basal water content throughout the ice sheet. In terms of the former (Fig. 12a), fast sliding occurs in grid points with a medium-size thickness (1.0-3.0 km), exhibiting a perfect correlation with water-saturated grid points (Fig. 12b).

In a somewhat more realistic approach to basal friction, we must consider the additional dependency on the effective pressure $\tau_b(u_b, N_{\text{eff}})$, thus triggering rapid ice streaming in temperate regions. Nevertheless, the assumption that ice streaming occurs in

all temperate grid points leads to an extremely low shear stress in the centre of the ice sheet (Fig. 6). For this reason, accounting for hydrological processes (e.g., the basal water content) appears to be fundamental to simulate Laurentide ice streams in accordance with geological reconstructions (Margold et al., 2015) and further yields ice-sheet volume and maximum elevation values closer to prior studies (Fig. 9). Besides, a water-dependent friction substantially considers the thermal state of the base, rather than just local dynamics. This implies a stress balance influenced by the geothermal heatflux as well as the frictional and

deformation heat contributions.

Overall, the simulated ice streams are numerically well-behaved and spatially constrained to lower troughs. In general, horizontal velocities reach an equilibrium value once the ice sheet has stabilized. However, global LIS variables as the total ice volume are highly sensitive to both the choice of friction law and the thermal coupling at the base.

## 5   Conclusions

We have simulated the LIS under LGM boundary conditions considering three basal friction scenarios of varying dynamic complexity and their consequences on the LIS ice streams, configuration, extent and volume.

First, in the purely mechanical friction formulation, we solely accounted for the force exerted by the weight of the ice column on a given grid point (overburden pressure). In this context, we considered three different dependencies of the basal shear stress on the sliding velocity: linear, purely plastic and regularized-Coulomb. Friction was thus independent of the thermal state of the base.

The LIS extent closely matches the reconstructed ICE-6G ice sheet, yet the volume appears to be slightly larger. For the linear case, this is a consequence of the absence of an active ice-stream network spatially constrained to low troughs that advects ice from the centre of the ice sheet to the margins. The surface elevation reflects a simple-domed ice sheet (except for the regularized-Coulomb scenario) resembling past results where the LIS deformable bedrock was ignored. Remarkably,

a fully-developed ice-stream network was simulated for a purely plastic and regularized-Coulomb formulation without any thermomechanical coupling requirements, yet the equilibrium ice volume appears to be slightly larger than previous reconstructions.

Hydrological processes were considered by coupling the basal friction to the thermal state of the base via the implementation of a water-dependent effective pressure formulation (Bueler and van Pelt, 2015). The simulated ice sheet also appears to be a multi-domed configuration with two relative maxima, yet the ice-stream structure strongly differs from the overburden approach for two reasons. First, the ice streams are spatially more restricted and second, the basal friction coefficient is generally higher for non-streaming regions. This approach yields the closest ice sheet volume to prior LIS reconstructions that also consider fast sliding in regions of a lubricated bed. These results support the hypothesis that hydrological processes are necessary to achieve physical realism in our simulations, specifically at aim of obtaining ice volume reconstruction similar to prior studies.

Notably, ice volume above flotation reached a constant equilibrium value for the all cases under consideration. Precise values are highly sensitive to thermomechanical coupling of the basal friction. The overburden case seems to overestimate the LIS volume compared to previous reconstructions. Nevertheless, significantly lower values are simulated when the thermal state of the base is accounted for, yet the particular coupling parametrization does not exhibit significant changes regarding ice volume nor total ice sheet extent. A water-dependent formulation yield volume and ice extent values substantially closer to prior studies.

Lastly, we can conclude that the most sophisticated scenario in this work (a thermomechanically coupled regularized-Coulomb basal friction) appears to be the closest reconstruction compared to prior ice-streams inventories. Future experiments shall focus on a more realistic basal hydrology, where conservative non-local processes (as the horizontal advection) are also resolved.

## Appendix A: The two-phase regression model

The two-phase linear regression model was studied by Hinkley (1969, 1971) and later also applied by Solow (1987). For our purpose, the underlying idea is to determine the *changepoint* in a given time series $y(t)$ to estimate the necessary length of the equilibration time in our simulations. Conceptually, the two-phase regression model assumes that there are two different behaviours in our data and these are captured by two independent linear functions (Eq. A.1). In the present study, these behaviours correspond to the transitory and stationary nature of the solutions respectively. The *changepoint* is thus defined as the abscissa of intersection that minimizes the residual sum of squares. Mathematically, we can write this model as:

$$y_i = \begin{cases} \alpha + \beta t_i, & i = 1, ..., r, \\ \gamma + \mu t_i, & i = r+1, ..., n, \end{cases} \tag{A.1}$$

where the abscissa of the intersection of these two regression lines reads:

$$t_c = \frac{\alpha - \gamma}{\mu - \beta} \tag{A.2}$$

and it is referred to as the changepoint.

Following Solow (1987), for our *changepoint* definition, we must ensure continuity of the underlying time series by imposing $t_c$ to lie in the interval $\mathfrak{I} \in (t_r, t_{r+1})$. Otherwise, the two-phase regression will include a discontinuity at $t_c$.

The approach thus aims at finding the estimate $t_c$. Since no closed form expression of $t_c$ is possible, the model given by A.1 is usually rewritten as:

$$y_i = \alpha + \beta t_i + \lambda \Omega_i(c) t_{i-c} + \varepsilon_i \tag{A.3}$$

where $\varepsilon_i$ is the error term, $\lambda = \mu - \beta$ and $\Omega_i(c)$ is given by:

$$\Omega_i(c) = \begin{cases} 0, & \text{if } i \le c, \\ 1, & \text{if } i > c. \end{cases} \tag{A.4}$$

Fixing a value of $c$, the modified model A.3 becomes a standard linear regression with two regressor variables: $t_i$ and $t_{i-c}$. Our problem is now reduced to finding $t_c$ so that its value minimizes the residual sum of squares (Fig. A1). For large datasets, Hinkley (1971) provides with a description of an efficient algorithm, though we simply apply a direct grid search given the dimensions of our time series.

Particularly, we used the ice volume above sea level as the regressand and performed the calculations aforementioned described. The vertical dashed line in Fig. 2 represent the abscissa of the changepoint $t_c$. Solow (1987) determines such value by minimizing the residual sum of squares RSS, though we will additionally compare these results with those given by maximizing the determination coefficient $R^2$ (Fig. A1). The values yielded by each method coincides.

*Author contributions.* Daniel Moreno Parada ran all the simulations, analysed the results and wrote the paper. All other authors contributed to analyse the results and writing the paper.

*Competing interests.* Alexander Robinson is an editor of The Cryosphere. The peer-review process was guided by an independent editor, and the authors have also no other competing interests to declare.

*Acknowledgements.* This research has been supported by the Spanish Ministry of Science and Innovation (project IceAge, grant no. PID2019-110714RA-100), the Ramón y Cajal Programme of the Spanish Ministry for Science, Innovation and Universities (grant no. RYC-2016-20587). This research is TiPES contribution no. 183 and has been supported by the European Union Horizon 2020 research and innovation program (grant no. 820970).

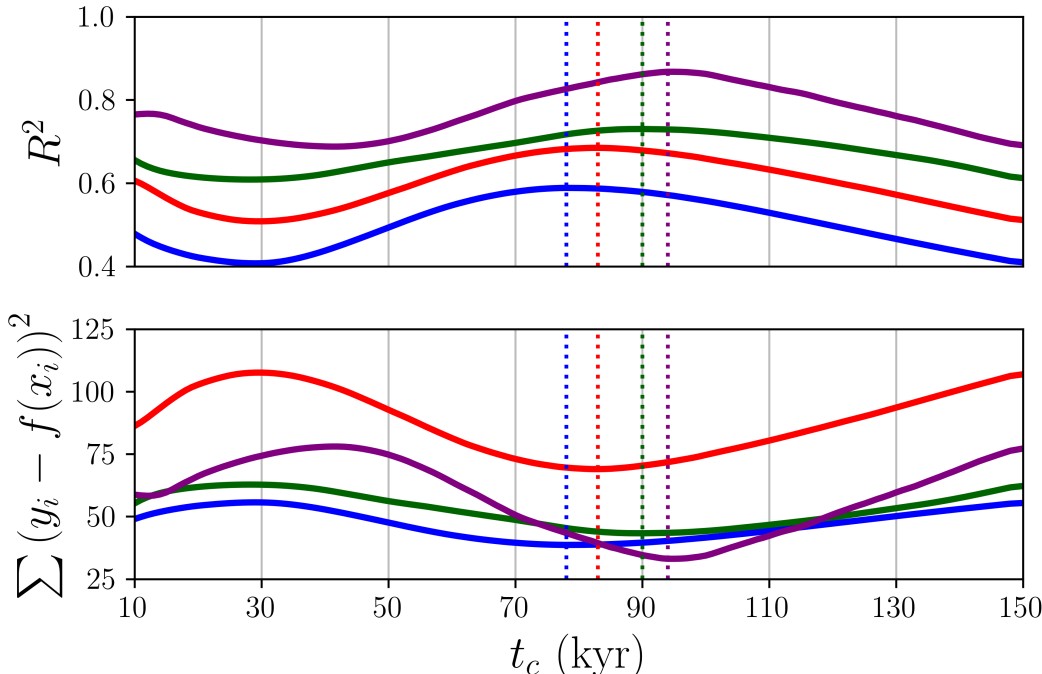

**Figure A1.** Determination coefficient $R^2$ (top panel) and residual sum of squares RSS (bottom panel) as a function of the fixed *changepoint* value taken. For each $t_c$ value, a standard linear regression (Eq. A.3) with two regressor variables is performed using the volume above sea level as a regressand. The vertical dashed lines correspond to the maximum and minimum values of $R^2$ and RSS respectively.

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
