# Peer review of "Simulating the Laurentide ice sheet of the Last Glacial Maximum"

_The Cryosphere, 2022_

## Author Comment (AC1)

**Authors final response to Julien Seguinot (RC1) TC-2022-215**

Daniel Moreno-Parada, Jorge Alvarez-Solas, Javier Blasco,
Marisa Montoya and Alexander Robinson.

February 13, 2023

**Summary**

The physical representation of glacier sliding (or basal friction) in numerical ice-sheet models remain a source of uncertainty and discrepancy. Some studies tune an empirical friction coefficient to velocity observations, while others use a more physics-based, albeit heavily parametrised, approach. In paleoglacier modelling the latter is typically preferred in the absence of velocity measurements.

This manuscript provide a strong case for the validity of this approach. The authors use basal friction physics of increasing complexity and assess modelled basal velocities. While most previous studies on the Laurentide ice sheet (and Cordilleran and Innuitian ice sheets) have focused on reproducing known ice-sheet extents, the authors introduce a novel approach using paleo-ice streams inventories as validation data instead. Expectedly, the different friction laws tested have little effect on ice-sheet extent, more effect on ice thickness, and a major effect on glacier velocities, demonstrating the value of complementary geomorphological datasets.

The authors make an assumption of steady-state. This is a logical choice, but it would deserve more discussion, I think (see general comment). Besides this point, I found the article very well redacted. The introduction is concise but exhaustive, and model physics are described in just enough detail to follow stand-alone without familiarity of the ice-sheet model Yelmo. Parts of the discussion could be shortened or improved (see specific comments) but overall, I enjoyed reading the paper and felt relief that the more advanced physics (water-dependent friction, similar to what I have used in my work) produced the most realistic velocity fields!

I fully support publication, and adress further comments directly to the authors.

The authors are deeply grateful for the thorough and constructive comments of the reviewer. This work will strongly benefit from them. We have responded to all comments below. Reviewer's comments are given in blue font whereas the author response reads in black.

**General comment**

My only general comment is about your steady-state assumption. I think this is a reasonable choice, but it also implies simplifications which I think should be discussed. Specifically:

- As mentioned in the introduction, the LGM ice-sheet configuration was likely transient. How representative is your modelled steady state of this transient configuration, what are the potential shortcomings? You mention several times that the simulations attain a steady state. How was this assessed? Are glacier velocities stable? Would you consider adding a line plot showing ice volume, area or thickness (maybe velocity or discharge if relevant)?

  Regarding the potential shortcomings of the modelled steady state. This is a simplification of course, but we wanted to study the impact of different friction laws under steady state conditions. We are planning to extent this research in the future with transient simulations.

  We have included line plots showing ice volume time series of our simulations (Fig. 2). Steady state has been assessed by a two-phase linear regression approach (following Hinkley, 1969, 1971; Solow, 1987). We have added Appendix A with a detailed description of the method. In fact, as the reviewer notes, glacier velocities, volume, area and thickness are stable. We employed ice volume above flotation time series as the regressand to determine the length of the spin-up and thus ensure an equilibrated state.

- Perhaps most importantly, how were timing inconsistencies taken into account in the validation step? Your target period is 21 ka. The LGM extent span a larger interval of 29-17 ka (numbers from your introduction), and ice streams inventories likely cover an even larger period including pre-LGM remnants and deglacial ice streams.

  Margold et al. (2015) already noted that no inferences on the timing of ice stream operation are possible because very few of the mapped ice streams have any chronological control. However, it is clear that the mapped ice stream tracks represent a time-transgressive imprint of evolving ice stream trajectories, i.e. they cannot have all operated at once. Nonetheless, some broad spatial patterns appear and we exploit this time-transgressive ice stream map to compare our simulations.

  In fact, the main idea behind the validation step was to obtain the main ice streams found in inventories. We then focus our attention on achieving a consistent steady state

and then compare our ice stream network with available inventories. Certain timing inconsistencies are thus inevitable, yet the the time-transgressive mapping remains as an appropriate reference for the simulated ice streams.

**Specific comments**

Some of this will be redundant with the above general comment.

- l. 120-122 "a minimum value $N = \delta P_0$": which value was chosen for $\delta$? This is an important parameter strongly influencing the results.

  We have inluded the chosen values following Bueler and van Pelt (2015). Namely, $\delta = 0.02$ in the present study.

- l. 195-197 "we tuned the ice-sheet model to obtain a fully-developed ice stream network (e.g., Margold et a l., 2015)": as this step influences results presented thereafter, I think it should be described in much more detail already here or in the methods. Is this a visual or quantitative comparison, please explain how it was done. Also, please explain precisely which data (implied by "e.g.") was used for validation.

  We have extended the description of this step and explain precisely which data was used for validation (see also our ice-streaming mask for a comparison with Fig. 2 in Margold et al., 2015).

  We mean a qualitative comparison. The text has been changed accordingly.

- l. 275 "a quantitative comparison": do you mean "qualitative"? If not, please explain how the data-model comparison was computed.

  Yes, we mean a qualitative comparison. The text has been changed accordingly.

- l. 296-302 "we find an interesting behaviour": I could not really understand the interesting behaviour and the role of SSA here.

  We are trying to understand the difference between Figs. 5c and 9a. We find non-monotonic minimum shear stress values in the low velocity regime ($u_b < 150$ m/yr) when the basal friction is coupled with thermodynamics. Nonetheless, all points taking part in this minimum shear stress region correspond to a fully drained till. Thus, any explicit water change does not explain the difference in behaviour. Presumably, we argue that those points with lowest $\tau_b$ cannot be reached given the new stress balance (i.e., the SSA equations) is changed if we account for $N_{\text{eff}}$. Since the SSA solution is non-local, the particular shape of $\tau_b(u_b)$ can be modified by a water-dependent effective pressure even for regions that are fully drained. This implicit effect would be a direct consequence of the non-local nature of the SSA solutions in regions where the water content remain constant.

  The paragraph has been modified and expanded for clarification.

- l. 361-364 "Figure 5 depicts... " I find that this paragraph largely repeats the results and is perhaps not necessary

  We have deleted this paragraph to avoid repetition.

- l. 365-368 "an interesting behaviour [...]": again, I think this is mostly just a description of the results (and I did not really understand this part).

  The explanation of this behaviour has been expanded in detail to foster comprehension and eludes repetition.

- l. 371-372 "For an idealized scenario [...] the behaviour imposed by Eq 7": this statement repeats the methods parts.

  We have deleted this statement and rephrase the following paragraph.

- l. 389-391 and 396-397 "a fully-developed ice-stream network was simulated [...] without any thermomechanical coupling requirements.", "hydrological processes are necessary to achieve physical realism". These two statements appear contradictory. It would help if you can be more explicit about the added benefits of water-dependent friction (also in the discussion part).

  Indeed, we have emphasized the fact the hydrological processes are necessary to obtain a closer volume to prior reconstructions. Besides, a water-dependent friction substantially considers the thermal state of the base, rather than just local dynamics. This implies a stress balance influenced by the geothermal heat flux as well as the frictional and deformation heat contributions.

- l. 398 "Notably, ice volume above flotation reached a constant equilibrium value for the three cases under consideration.": This conclusion is not supported by the results. Please remove, or add a figure.

  The reviewer is right. We have further performed a two-phase linear regression following Solow (1987) to rigorously quantify the equilibration time and further included the volume times series (Fig . 2). This additionally proves that our simulations have no drift. An appendix will be included with a description of the calculation and a figure supporting the statement will be added.

**Technical corrections**

In the title, and throughout the text, you use "Laurentide ice sheet" to refer to what many paleoglaciologists would call the "North American ice sheets" or "North American ice-sheet complex" (including the Laurentide, Cordilleran, and Innuitian ice sheets). In paleoclimate literature though, "Laurentide" generally includes all of North America. Of course, please use the terms you find most appropriate to your target audience. But I wanted to let you know that to some readers, it almost sounds like you are downplaying your results and it could be worth clearly mentioning the scope of your study somewhere early on.

We thank the reviewer for their comment. We have reconsidered the terminology to avoid misunderstanding among certain readers. "Laurentide" has been kept in our title, though we have included a brief clarification on our terminology in the introduction.

- l. 38-42, 343-345, Table 2, etc. While you give s.l.e. estimates in the intro, you use ice volumes in cubic kilometres in the rest of the text. Personally I found the s.l.e. values very useful as they are easier to compare to each other and to the known global total of ca. 120 metres, and would recommend sticking to them throughout the text.

  For the introduction, we kept the same units as they were given in the corresponding reference. Nevertheless, our results and discussion sections employ cubic metres to facilitate comparison with prior compilation of published estimates of the LIS (e.g., Table 1, Stokes, 2017). To foster comparison with the present work, we have also included a figure with a comparison among LIS reconstructed ice sheet extent, max. elevation and volume.

- l. 98 "Here, *Q* is the net heat flow": *Q* is missing from the preceding equation. We thank the reviewer, this has been fixed.

- l. 99-100 "Bueler and van Pelt (2015)": brackets are missing around the reference. This has been fixed. Thank you.

- l. 218 "coulomb" : should have a capital C.

  We thank the reviewer, this has been fixed.

- l. 267, etc "Thermomechanical coupling": this phrase is used throughout the manuscript, but it can be a bit counter-intuitive. In glaciological literature "thermomechanical coupling" is often used in e.g. "thermomechanically coupled SIA, to describe the thermal feedback on ice rheology, which is included in all your simulations. I would recommend using a different phrase or introduce regular reminders of what you mean with "thermomechanical coupling", otherwise a reader only skimming through figure captions and conclusions could get it wrong.

  We have included reminders that the thermomechanical coupling refers to the basal friction. We thank the reviewer the comment.

- l. 280 "Fig 2f as compared to Fig 2": should this be "Fig 8a as compared to Fig 2f"?

  Indeed, the reviewer is right.

- l. 296 "this idea": this is a result, more than an idea. :-)

  We thank the reviewer, this has been fixed.

- l. 313-314 "Quantitatively": is it "qualitatively"? Same commend as above.

  This has been fixed.

- l. 335 "a purely thermodynamic perspective": I can't understand what you mean with these words.

  We meant that the thermal state of the ice-sheet base is sensitive to the particular $\tau_b(u_b)$ choice (see Fig. 3). The text has been changed accordingly for clarification.

- l. 341 "ICE-6": ICE-6G

  We thank the reviewer, this typo has been fixed.

- l. 356-357 "surface elevation...our 4.5 kilometres thickness.": in this statement you compare surface altitude to thickness, which are different things. Besides, which run does the second number refer to?

  The reviewer is right, it does not allow for a one-to-one comparison. In fact, values from Boulton et al (1985) and Fisher et al. (1985) are given as maximum elevation above present sea level. We have also included their volume reconstructions to give a more accurate idea of the differences among results.

  Figure 9 further clarifies the statement and provides a neat comparison among prior studies and our current simulations, thus putting into perspective results presented in the latter.

Apologies for the very late review, happy celebrations, and good luck with your future endeavours!

We are grateful for the thorough comments of the reviewer. The current work will firmly benefit from them.

---

## Author Comment (AC2)

**Authors final response to Niall Gandy (RC2)**
**TC-2022-215**

Daniel Moreno-Parada, Jorge Alvarez-Solas, Javier Blasco,
Marisa Montoya and Alexander Robinson.

February 13, 2023

The authors are deeply grateful for the elaborated and constructive comments of the reviewer. This work will firmly benefit from them. We have responded to all comments below. Reviewer's comments are given in blue font whereas the author response reads in black.

**Summary**

Moreno et al. present a series of simulations of the North American Ice Sheets, exploring the resulting ice sheet volume, area, and velocity pattern from varying the ice sheet sliding law. While the results show only limited variation in the ice sheet volume and area, the ice sheet velocity pattern is sensitive to the sliding law used.

This manuscript is well presented, with clear text and figures. Most importantly, the work as been clearly and comprehensively described, and the results are presented and discussed in good detail. I recommend that the manuscript is published following minor corrections/clarifications (detailed below).

I hope you have a good Christmas break, and I hope to reading a revised or published version of the manuscript in the new year.

**General Comments**

A direct visual comparison between the ice stream dataset from Margold et al and your results here would be useful. Essentially, it saves the reader flicking between browser tabs,

and I expect it would show clearly the match you have described in the text. Within this it would be good to discuss the potentially transient nature of some ice streams, and how this might effect the empirical mapping, your modelling results, and the comparison between the two.

Margold et al. (2015) data is not publicly available. A visual comparison with previous LIS reconstructions is already provided in all 2D plots by the ice-sheet extent taken from ICE6G (Peltier et al., 2015). Particularly, Margold et al. (2015) provided with an ice stream inventory from geological reconstructions and does not allow for a quantitative comparison as such. Available information about individual ice streams of the Laurentide Ice Sheet can be found in the Supplementary material of Margold et al. (2015).

I think it is reasonable that you have run your simulations to equilibrium, but it will probably have an effect on your results, given ice stream sensitivity to climate forcing. This should be discussed in the text.

Indeed, transient simulations may have an impact given ice stream sensitivity, though minor implications on the results of the present work. We must keep in mind that we are comparing our ice streams to prior inventories (e.g., Margold et al., 2015). It is clear that their mapped ice stream tracks represent a time-transgressive imprint of evolving ice stream trajectories, i.e. they can not have all operated at once. Nonetheless, some broad spatial patterns appear and we further exploit this fact to compare our simulations. Potential timing inconsistencies are thus inevitable, though the time-transgressive inventory remains as an appropriate reference for the simulated ice streams.

The discussion section have been expanded to account for this simplification and transient simulations are in fact in the scope for future work.

**Minor points**

- Title: While I would often refer to the ice simulated here as the "Laurentide", more formally I would opt for "North American Ice Sheets". Laurentide is neater, North American Ice Sheets is clearer. If you stick with Laurentide consider a very brief mention in the Introduction.

  We have kept "Laurentide" in our title, though we have included a brief clarification on our terminology in the introduction. We thank the reviewer for this comment.

- Ln 25: "Strictly speaking"... This sentence isn't clear to me, please rephrase

  This statement has been rephrased.

- Ln 28: References for the initial assertion? Perhaps Calov et al., 2002, Tarasov and Peltier, 2004, or others?

  We have included additional references. We thank the reviewer.

- Ln 36: Extension > extent – and other uses later in the manuscript

  This has been fixed throughout the text.

- Ln 36: "largely differ" > "differ largely"

  The manuscript has been corrected accordingly.

- Ln 44: If the variable ice thickness is through a surging/instability mechanism say this explicitly. This paragraphs touches on the idea that ice stream instabilities could significantly influence the ice sheet configuration, but more detail/references would be appreciated.

  Additional references have been included upon the idea that ice streams instabilities could influence the ice sheet configuration.

- Ln 71: Please provide some further justification for these parameter values.

  In the current study, the enhancement factor is treated as a tuning parameter. Laboratory experiments provide the basis for estimating such parameter (e.g., Russell-Head and Budd, 1979; Baker, 1981,1982). More recently, Budd and Jacka (1989) and Jacka and Maccagnan (1984) have suggested enhancement factors up to 3. We employ typical values found in Ma et al. (2010). An example of spanned parameter range can be found in ISMIP6 (Seroussi et al., 2020).

- Ln 95: It's pretty typical to ignore horizontal water transport, but not always (e.g. Gowan et al., 2018). It's worth justifying this simplification.

  Considering horizontal advection is indeed a more sophisticated description. However, our simplification is justified since we assumed till properties similar to Tulaczyk et al. (2000). The hydraulic diffusion coefficient shares the same order of magnitude $c_v \sim 10^{-8}~m^2/s$, hence horizontal advection becomes negligible compared to the local basal mass balance.

- Ln 104: What happens to excess water beyond the 2 m limit? Does it accumulate but is ignored, or disappear?

  This is in fact one of the caveats of the local non-conserving approach. Once the 2 m limit is reached, any additional water production disappears. Water is therefore not strictly conserved. A better representation of subglacial hydrology is in the scope of future work.

- Ln 173: By averaging 11 PMIP simulations you remove the consistent climatology provided by a single model. Is this important?

In this study, we want constant boundary conditions rather than a time-dependent forcing for our simulations. Thus, taking the average among climatologies gives us a more robust boundary condition and smoothes potential peculiarities of each General Circulation Model.

Particularly, some authors have argued that: *"Just as the mean of n uncorrelated random variables with variance 1 should have variance $1/n$, we should expect that the ensemble mean of independent models defined in this way would (a) perform better than any individual simulation, and (b) asymptotically converge to zero error as the size of the ensemble of independent models (with zero error correlation) increases."* (Abramowitz et al., 2018).

If we were to study transient behaviour, consistent climatologies provided by each model should be considered independently.

- Ln 173: Are all 11 PMIP simulations using the same ice sheet reconstruction?

Yes, as described in their experimental design (`https://pmip3.lsce.ipsl.fr/`), the ice sheet extent and related changes in topography is prescribed and provided for the *21 ka - Last Glacial Maximum experiment* (see PMIP 3-CMIP 5 Experimental Design).

- Ln 196: As your climate forcing in Figure 1a and b inherently contains a previous ice sheet reconstruction which broadly matches the empirical reconstruction, how surprising is it that your simulated extents are okay?

Whilst ice extent is implicitly contained in our climatic forcing, the simulated LIS extension is still highly sensitive to the ice-sheet model parameter choice. In the end it is a question of surface mass balance, determined by the climatology and affected by the extent (and more notably the elevation) and dynamics, which are very much dependent on model parameters. For the same climatic forcing, we could obtain an ice sheet that would largely differ from prior reconstructions solely by employing a different parameter space. The interesting result is that we did not tune our model to match a certain volume/extent value, but rather to develop an ice stream network comparable with existing inventories.

Moreover, in order to avoid any inertia of the model to evolve towards the inherent previous ice extent, we further apply a lapse rate factor correction of the temperature and precipitation PMIP3 forcing fields as a function of the local surface elevation.

- Figure 2: It would be good to see a direct visual comparison to the Margold ice stream reconstruction. It would also like to see one section of the ice sheet in more detail to show the nature of ice streaming at the margin. Maybe there could be a separate plot of the Hudson Bay and surrounding ice streams?

The main focus of the present work is on the general ice stream network configuration of the LIS rather than on certain located areas. Nevertheless, following the comment made on Line 256, we have expanded Table 2 so as to account for equilibrium fluxes and mean/min/stdev velocities. Additionally, there is a new figure (Fig. 11) that

complements the description in the text and captures the changes in the probability density function of the basal temperature and the sliding velocity.

- Ln 256: A table summarising key statistics of Linear, plastic, and coulomb simulations might be helpful to compliment to description in the text. A quick lookup for the equilibrium fluxes, mean/min/stdev velocities would be appreciated, perhaps an extension to Table 2?

  We have included an extension to Table 2 with equilibrium fluxes and mean/min/stdev velocities as a complement the description in the manuscript. Moreover, we have included an additional figure (Fig. 11) that complements the description in the text and captures the changes in the probability density function of the basal temperature and the sliding velocity.

- Ln 275: Quantitative or Qualitative?

  Qualitative. This typo has been fixed.

- Figure 6: This is a very useful figure. There seems to be an edge effect stripe in panel b and c (around 100m Hice). Do you know what is causing this?

  It is presumably a consequence of the minimum ice thickness considered by the model. For these simulations, we set such value at 100 metres, so that a grid cell with a smaller amount of ice is neglected in the following time step.

- Figure 7: This figure is good at showing the model's behaviour in general, but the visual comparison between sliding laws is tricky? Perhaps you could experiment with plotting curves from multiple simulations on the same panel.

  We first plotted multiple simulations in the same panel, yet it worsened the visualization as certain lines are quite close to each other. We thank the reviewer for this comment.

- Ln 341: ICE-6G

  This typo has been fixed.

- Ln 410: Where will the data from the simulations be available?

  Data will be stored in a Zenodo repository.

---

## Author Response (AR2)

**Authors final response**
**TC-2022-215**

Daniel Moreno-Parada, Jorge Alvarez-Solas, Javier Blasco,
Marisa Montoya and Alexander Robinson.

March 29, 2023

The authors are truly grateful for the elaborated and constructive comments of the editor.
This work has firmly benefit from them. We have updated the manuscript accordingly and
made sure that all subplot panels are merged into one single figure.